# Visual Affective Stimulus Database: A Validated Set of Short Videos

**DOI:** 10.3390/bs12050137

**Published:** 2022-05-09

**Authors:** Qiuhong Li, Yiran Zhao, Bingyan Gong, Ruyue Li, Yinqiao Wang, Xinyuan Yan, Chao Wu

**Affiliations:** 1School of Nursing, Peking University, Beijing 100191, China; qiuhong_li@pku.edu.cn (Q.L.); 15650715058@163.com (Y.Z.); bingyangong15@163.com (B.G.); 18838956367@163.com (R.L.); 2Department of Machine Intelligence, Shandong University, Jinan 250100, China; infamywong@gmail.com; 3School of Computing, University of Utah, Salt Lake City, UT 84074, USA; xyan@cs.utah.edu

**Keywords:** emotion, emotion measurement, affective visual stimuli, affective short-video library, psychometrics

## Abstract

Two hundred and ninety-nine videos representing four categories (people, animals, objects, and scenes) were standardized using Adobe Premiere Pro CC 2018, with a unified duration of 3 s, a resolution of 1080 pixels/inch, and a size of 1920 × 1080 mm^2^. One-hundred and sixteen participants (mean age 22.60 ± 2.06 years; 51 males) assessed the videos by scoring, on a self-reported 9-point scale, three dimensions of emotion: valence, arousal, and dominance. The content was attributed a specific valence (positive, neutral, or negative) if more than 60% of the participants identified it with an emotion category. Results: In total, 242 short videos, including 112 positive videos, 47 neutral videos, and 83 negative videos, were retained in the video stimuli database. In the three-dimensional degree of emotion, the distribution relationship between them conformed to the fundamental characteristics of emotion. The internal consistency reliability coefficient for valence, arousal, and dominance was 0.968, 0.984, and 0.970. The internal consistency reliability of the emotional dimensions for people and faces, animals, objects, and scenes ranged between 0.799 and 0.968. Conclusions: The emotion short-video system contains multi-scene dynamic stimuli with good reliability and scoring distribution and is applicable in emotion and emotion-related research.

## 1. Introduction

Emotion has a significant impact on individuals’ survival and adaptation. Emotions can influence the cognitive processes of humans by exerting an effect on subjective feeling, conflict handling, and decision making [1,2,3]. For instance, implementing emotion regulation strategies is conducive to executing cognitive control tasks [4]. Meanwhile, a variety of mental disorders (e.g., schizophrenia and autism) have emotion perception abnormalities [5], and dysregulation of emotion is a risk factor for the relapse of many psychiatric disorders, such as obsessive-compulsive disorder, schizophrenia, and mania [6]. Emotions arise from the cognitive evaluation of a specific situation [7], and in the study of emotion, one critical step is the induction of a particular emotion in individuals, with a manifest effect.

Generally, emotional stimulus databases include visual, auditory, olfactory, and tactile materials. Visual stimuli are the most commonly used materials in psychological research because they are intuitive to feel, easy to use, and accurate when measuring the start and the end of stimuli. The International Affective Picture System (IAPS) is the world’s most widely used database [8]. In the IAPS, pictures are assessed by the three dimensions of valence, arousal, and dominance. Multiple studies have demonstrated the reliability of the IAPS as regards the triggering of emotions and affective responses [9,10,11]. Chinese researchers found that Chinese participants responded differently to certain emotional materials in comparison to people in western countries, which may be due to cultural and ethnic differences [12,13]. Consequently, a study developed the Chinese Affective Picture System (CAPS) by introducing pictures with Oriental characteristics based on the IAPS [14].

Previous studies pointed out that the emotional intensity caused by dynamic expressions is higher than that of static ones [15]. Presently, several video emotional stimulus databases are available. For example, some video libraries containing facial expressions and sounds have been developed and used worldwide [5,16,17,18,19,20]. In these face emotion video libraries, there are datasets containing images of dynamic emotional faces generated by professional actors, such as the Geneva Multimodal Emotion Portrayal (GEMEP) corpus [16]. A library, the Child Emotion Facial Expression Set (ChildEFES), contains dynamic face expressions recorded by children [17]. In addition, some emotional videos are from film clips and include characters and environmental scenes [21,22]. For example, in 2010, Xu et al. created a Chinese Affective Video System [21] that contains 30 video clips taken from public domestic films or non-commercial videos. However, the results showed that the emotional intensity of participants watching those videos was significantly affected by whether they had seen the clips before, and that the database has limitations because of the small number of videos it contains.

In real life, the generation of emotion mainly relates to dynamic simulation. Dynamic emotion videos [23] and emotion neural computing models or brain-like emotion recognition algorithms [24,25] have been widely used in social scene analysis. The demand for stimuli with natural ecological validity in emotion recognition research is increasing, but due to the limitations of static stimuli and of the content of previous video stimulus databases (i.e., the abovementioned databases include facial expressions or upper body movements only or clips from non-commercial videos), it is necessary to expand the scope of dynamic visual stimulus materials to multiple categories of real natural scenes and character scenes in the research of emotion. Thus, in this study, to fill this research gap and provide more optional materials for researchers, we aimed to construct a short-video database containing various categories, including people, animals, objects, and natural and social scenes. We then tested the database among college students.

## 2. Materials and Methods

### 2.1. Participants

In this study, emotions were rated according to three dimensions (valence, arousal, and dominance), and the short videos were classified into four categories (people, animals, objects, and scenes). The sample size was calculated by referring to the method specified in exploratory factor analysis and is generally 5–10 times the number of items on the scale used [26]. Therefore, it had to be [12 × 5, 12 × 10], i.e., the sample size ranged from 60 to 120 participants.

We used the convenience sampling method to recruit participants. One hundred and seventy-seven college students took part in the study, and four dropped out midway. Of the 173 participants who completed the questionnaires and video assessments, after removing 57 participants who had scores ≥14 on the Baker Depression Scale [27] or ≥10 on the Generalized Anxiety Disorder Scale [28], we included in the analysis the remaining 116 students. Among them, 51 were males, and 65 were females; they were aged 18 to 27 years, with an average of 22.60 ± 2.06 years. There was no significant difference in age between genders (t = −0.110, *p* = 0.912). This study was reviewed and approved by the Medical Ethics Committee of Peking University (IRB00001052-21046). All participants were voluntary and signed an informed consent before the study. After completing all the experiments, the participants obtained a fee.

### 2.2. Video Materials

The authors took the videos in various places and locations with iPhone or Huawei mobile phones in the resolution of 4K frames per second, and a total of 575 clips were obtained. Among them, 299 clips with explicit content and precise emotional expressions were selected through preliminary screening by four researchers. Those excluded were mainly because of their vague meaning or because they were unclear images or shaky camera shots. Using Adobe Premiere Pro CC 2018, the 299 videos were processed into 3 s videos with a resolution of 1080 pixels per inch and a size of 1920 × 1080 mm^2^.

After the selection procedure, the image classification method proposed by Colden et al. [29] was employed to categorize the videos into four categories, facilitating the stimulus selection for future studies. Videos of the “people” category were those showing live, injured, or dead human bodies or isolated parts of the human body (some videos were shot with body props). This category contained facial information, with at least the eyes or the mouth region visible. Given the distinctions between pictures and videos, the “people” videos had specific actions performed by humans. Videos of the “animal” category showed live or dead animals (some videos were shot with animal props). Some videos in this category contained certain parts of the human body (not faces) in the background (e.g., a person with an animal in his hands). The “object” category videos showed diverse visible objects, such as food or vehicles, depicted without humans or animals. Finally, the “scene” category videos contained a wide range of natural and artificial landscapes, panoramas, or terrains. The four researchers categorized the 299 videos independently and classified approximately 99% of the videos into the same category. The researchers resolved any disagreement through discussion according to the descriptions above.

### 2.3. Sociodemographic Variables and Self-Administered Questionnaire

The biological sex of female vs. male was coded as 1 vs. 0. Education level was categorized into two subgroups (college = 0, postgraduate or above = 1). Marital status was defined as married (coded as 1) or unmarried (coded as 0). Recent life events were investigated by the following self-made questions: 1. Have you lost important relatives or friends in the last month? 2. Have you divorced or felt lovelorn in the last month? Have you encountered any unacceptable setbacks in your studies or career in the last month? Have you experienced any events other than the above problems in the last month that have significantly affected your mood? If the answer to the above question was yes, the score was 1; if the answer was no, the score was 0.

#### 2.3.1. Beck Depression Inventory-II in Chinese Version (BDI-II-C)

The Beck Depression Inventory [30] was used to assess the severity of participants’ depressive symptoms in the past two weeks. The score of each item ranges from 0 to 3, and the total score ranges from 0 to 63. The original scale provides score boundaries: a total score of 0–13 indicates no depression, a score of 14–19 mild depression, a score of 20–28 moderate depression, and a score of 29–63 severe depression. This study used the version translated by Wang et al. [31], which had a Cronbach alpha coefficient of 0.85 among first-year college students. The sensitivity of screening for all depressive disorders was 99% at a cut-off score of ≥14 in Chinese adolescents [32].

#### 2.3.2. Generalized Anxiety Disorder Scale (7-Item Generalized Anxiety Disorder Scale, GAD-7)

The GAD-7 [33] was used to assess participants’ anxiety severity over two weeks, with seven entries, each scored from 0 to 3, and a total score ranging from 0 to 21. The anxiety status is divided into four levels according to the total score, with 0–4 indicating no GAD, 5–9 mild anxiety, 10–14 moderate anxiety, and 15–21 severe anxiety [34]. In this study, we used the version translated by He et al. [35], which had a Cronbach’s α coefficient of 0.898 and a retest reliability coefficient of 0.856 in outpatient clinics in general hospitals. The sensitivity and specificity were 86.2% and 95.5% when the cut-off score was 10, indicating the scale’s good reliability and validity [28].

#### 2.3.3. Emotion Rating Scale

Before the experiment, the participants were instructed to practice rating the videos. After watching the videos, the participants were asked to score their emotional responses (valence, arousal, and dominance) on a 9-point scale. On the valence scale, 1 represents very negative emotions or feelings (unpleasant, e.g., disgusted, angry, or sad), 5 means no emotional change (neutral), and 9 indicates very positive (pleasant, e.g., happy, interested, or satisfied) emotions. On the arousal scale, 1 point indicates high passivity (e.g., drowsy, sluggish, or unable to lift the spirits); 5 points mean no emotional change (neutral); 9 points indicate high activity (e.g., in high spirits, excited, alert, or stimulated). On the dominance scale, 1 point indicates a controlled state (e.g., involuntary, under control, awe-stricken, subdued, or repressed), 5 points mean no emotional change (neutral), and 9 points indicate being in control of the situation (e.g., initiative, influential, powerful).

### 2.4. Stimuli Presentation and Online Experiment Procedure

This study was conducted in the spring of 2020. Due to the global COVID-19 pandemic at that time, we chose to perform our research online. We developed a website using JavaScript (version 2018) and nodeJS (version 4.4.3) for this experiment and stored visual clips on the cloud server. A user could log into the website with a unique assigned username and password to start the experiment. Participants were instructed to watch each video only once and score the three emotional dimensions in the slider at the bottom of the web page after each display. Only after watching a video could the participants rate it. The rating scales remained available to the participant until they completed valence, arousal, and dominance ratings. The individual scoring time of each video was up to the participant, accounting for 1.5 to 2 h in total. The participants were told not to think and to rate according to their feelings at that moment. After each rating, by clicking “Next,” the following video appeared. The playing order was random for each participant.

### 2.5. Statistical Analysis

The statistical analyses were performed using the software SPSS 22.0. We first used descriptive statistics to show the sample characteristics under various conditions within-group or between-group, then independent sample *t*-tests to compare the mean differences of variables between gender groups, and finally scatter charts and correlation analyses to show the distribution characteristics between two of the three dimensions (valence, arousal, and dominance) of emotion under various conditions.

## 3. Results

### 3.1. Rating of Emotional Videos

On the valence dimension, a video was determined to induce a negative emotion when it scored less than 5, a neutral emotion when its score was equal to 5, and a positive emotion when its score was over 5. When videos are categorized according to the valence dimension, the recognition rate is expected to be higher than 60.0% [36,37], that is to say, 60 out of 100 participants rated the video as the target emotion. For this reason, of the 299 videos, 57 videos whose ratings were lower than 60%were removed. Among the 242 selected videos, the videos with positive, neutral, or negative properties were 112, 47, and 83, respectively (Appendix A). The recognition rate and category information for the 242 out of 299 videos are listed in Appendix A. The average (across participants) score of each of the 242 videos for each emotional dimension (valence, arousal, and dominance) is shown in Appendix A.

### 3.2. Internal Consistency

The internal consistency reliability coefficient (Cronbach’s Alpha) of 116 participants for the evaluation of 242 videos by of participants for the dimensions of valence, arousal, and dominance was 0.968, 0.984, and 0.970, respectively, and the internal consistency was higher than 0.92 across the positive, neutral, and negative categories (Table 1). For each emotional dimension, the internal consistency reliability coefficient of the videos in the four categories was higher than 0.799 (Table 2), indicating high reliability in each emotional dimension and classification [38].

### 3.3. Ratings for Emotional Dimensions and Gender Differences

The scores in the valence and dominance dimensions were widely distributed, with larger range values (the difference between the Max and the Min value); in the arousal dimension, the scores tended to converge and meet at more than 4.5, with a narrow standard deviation. There were no significant sex differences in the three emotional dimensions. Compared to the male participants, the female participants did not show any sensitivity in recognizing negative, positive, or neutral videos (all *p* values > 0.1).

### 3.4. Scatter Plots of Rating Distribution for Emotional Dimensions

We made several pairwise correlation analyses between the scores of the three emotion dimensions of valence, arousal, and dominance, obtaining the correlation distribution in Figure 1. When there was a nonlinear relationship in the scatter diagram, we conducted three-dimensional correlation analysis for positive and negative videos, separately. Each single dot represents the average rating for one video on a two-dimensional scale, averaged across participants.

The scatter plot shown in the upper panel of Figure 1 illustrates the distribution of the short emotional videos in the affective space created by the valence and arousal dimensions. The resulting distribution pattern resembles a “V” opening upward, with high arousal in high/low valence areas, and neutral arousal in the neutral valence area (positive videos: r = 0.875, *p* < 0.001; negative videos: r = −0.659, *p* < 0.001). There were no sex differences in the distribution. When the videos were divided into the four categories of humans, animals, objects, and scenes, the inequality of the distribution became apparent. In the people and animal categories, a dense concentration of ratings was in the positive or negative emotion area; in the object category, the ratings were uniformly distributed; in the scene category, the rating scores were mainly in the neutral or positive areas. In the middle panel of Figure 1, valence and dominance were positively correlated (r = 0.998, *p* < 0.001). The rating of valence increased as that of the dominance dimension. The human and animal categories were mainly in the high and low dominant areas, objects were uniformly distributed, and scenes were in the highly controlled area. The distribution pattern in the bottom panel of Figure 1 forms a “boomerang,” with one wing extending toward the low dominance/high arousal area, another wing extending toward the high dominance/arousal area, and the connecting angle of the boomerang located in the neutral dominance and low arousal area. The level of being aroused increased as the intensity of dominance went either upward or downward (positive videos: r = 0.876, *p* < 0.001; negative videos: r = −0.753, *p* < 0.001).

## 4. Discussion

This study established an emotional stimulus material system composed of dynamic short videos drawn from real life with events and scenes. The valence of the videos was positive, neutral, or negative. The obtained rating patterns from the valence, arousal, and dominance dimensions corresponded with the distribution of basic emotions and were similar to the distribution of the IAPS, indicating a high validity of the video stimuli system. The reliability coefficients for valence, arousal, and dominance were 0.968, 0.984, and 0.970. The internal consistency of the videos in different categories ranged from 0.799 to 0.968.

The short videos in this study have two advantages, i.e., they can facilitate future research and expand researchers’ options. Each stimulus of the emotional stimulus material system expresses a precise and dynamic meaning of emotions and appeared closer to the actual situation of emotion generation. Another advantage is that we included four categories of videos from diverse sources, i.e., people (including human faces), animals, objects, and scenes. Positive stimuli contained natural sceneries, artificial landscapes, lovely animals, attractive food, smiling faces, interacting family scenes, and playing children. Neutral videos covered everyday objects, buildings, ordinary natural sceneries, and everyday facilities. Negative videos included those with bloody scenes (shot on props), ghosts (played by researchers), crying faces, disasters (e.g., fires, hospitals), garbage-related scenes (e.g., landfill, trash cans), insects, etc.

The scatter plots showed that arousal and dominance increased as pleasantness grew, rosed up and decreased as the unpleasantness increased, and remained neutral when valence was near the neutral area. These findings are consistent with those of the IAPS and CAPS [14], indicating that the emotions induced by this emotional stimulus material system in short videos conform to the fundamental law of emotion distribution.

The present stimulus system obtained similar maximum values in the three dimensions of valence, arousal, and dominance as those of the CAPS [14], with ratings for the valence of all stimuli not higher than 8. One explanation is a lack of high valence/high arousal materials in the two databases. Another explanation is that many Chinese college students maintain conventional conservative thinking and spare reservations in unforeseen circumstances, therefore rarely giving extreme high or low scores. A study [13] compared the rating differences between American students and Chinese students on IAPS and found that, compared to Chinese students, American students usually scored higher on the valence dimension. Moreover, apart from the distinctions caused by pictures with noticeable cultural differences, some highly enjoyable scenery also induced a lower level of pleasure among Chinese students. The findings may manifest the “conservative marking” of Chinese students. The minimum and mean rating values for our video stimuli were higher than those of the arousal scale of CAPS due to the natural differences between pictures and videos. Dynamic videos can make raters more aroused and less sleepy or feel dull than static pictures.

In this study, gender differences were not significant for the three emotional dimensions. However, Yi et al. study [39] found that women responded more negatively than men when facing negative video stimuli. These sex differences may be due to differences in brain electrophysiological responses between men and women during emotional processing. For example, when responding to negative images, females generally have a decreased frontal latency (mainly on the right side). However, such a phenomenon did not appear in this study using short emotional videos. Sex differences in evaluating emotional stimuli may be affected by many factors, such as childhood maltreatment experiences [40] and personality [41]. Future research needs to include more variables related to emotion to explore the gender differences in perceiving dynamic emotion stimuli.

The strengths of this study include: (1) the videos extended standard tools like CAPS by adding the dynamic dimension, (2) the exclusion of individuals with possible clinical depression or anxiety avoided the impact of these two conditions on the emotion rating of the videos; (3) randomizing the order of video playing and balancing the participants’ gender ensured the standardization of the research; (4) from the perspective of the application prospect of the video library, future studies may assess the application of the video library in other age groups and patient populations. Further, emotion videos have been used in recent social cognition studies, such as in the areas of visually mediated empathy and somatosensory vicariousness and their interaction with neural activities [42,43]. Dynamic visual affective stimuli are also valuable for emotional neutral computing models, such as recognizing personality [44] or distinguishing emotions [45] through emotional video-induced EEG signals. These application prospects imply that the visual emotion video library tested in this study, which has a standardized classification of the experimental stimuli, might help improve the design of human affective-oriented experiments and the accuracy of neural computing predictive models.

This study also has some limitations. First, the total number and categories of the short-video stimuli should be expanded, especially those in the high arousal/valence areas. Second, in the video evaluation, there was no test for the classification of complex emotions represented in the video, such as to judge whether the positive emotions reflected in a video were joy or amusement and whether the negative emotions reflected in a video were sadness, anger, or fear. Future research can further improve this aspect and supplement videos as needed. Third, some social and natural scenes in the videos may have Oriental cultural characteristics, such as garden sceneries and group dance, so their cross-cultural applicability needs to be further verified.

## 5. Conclusions

This study established an emotion short-video system containing dynamic stimuli with diverse scenes in multiple categories and demonstrated its high reliability and validity. The corpus can supplement other emotional visual materials in related research areas.

## Figures and Tables

**Figure 1 behavsci-12-00137-f001:**
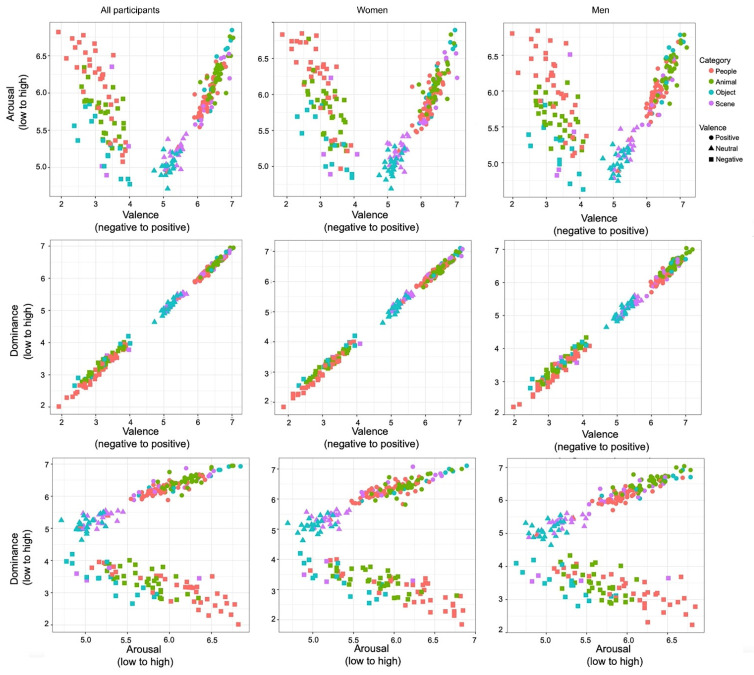
Distribution of emotion dimensions. Upper panel: Emotional ratings of arousal (*y*-axis) and valence (*x*-axis) for each category by all participants, women, and men. Middle panel: Emotional ratings of dominance (*y*-axis) and valence (*x*-axis) for each category by all participants, women, and men. Lower panel: Emotional ratings of dominance (*y*-axis) and arousal (*x*-axis) for each category by all participants, women, and men. Each single dot represents the average rating (across participants) of a video on a two-dimensional scale.

**Table 1 behavsci-12-00137-t001:** Internal consistency reliability for three emotion dimensions and in relation to their positive, neutral, and negative valence.

	Valence	Arousal	Dominance
Total (*n* = 242)	0.968	0.984	0.970
Positive (*n* = 112)	0.987	0.987	0.987
Neutral (*n* = 47)	0.923	0.937	0.920
Negative (*n* = 83)	0.973	0.982	0.978

**Table 2 behavsci-12-00137-t002:** Internal consistency reliability for each emotion dimension across categories.

	Valence	Arousal	Dominance
People (*n* = 84)	0.923	0.968	0.931
Animals (*n* = 68)	0.932	0.966	0.931
Objects (*n* = 47)	0.799	0.879	0.800
Scene (*n* = 43)	0.933	0.936	0.938

## Data Availability

A video sample can be found at https://osf.io/rhuzy/files/ (accessed on 1 May 2022).

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
