# Peer review of "Visual Affective Stimulus Database: A Validated Set of Short Videos"

_behavsci, 2022, doi:10.3390/bs12050137_

Round 1
Reviewer 1 Report
The research work and contribution of authors are to provide short videos for visual emotion research and build the corresponding database, which has certain value for the construction and testing of positive and negative samples in subsequent research.
My suggestions are as follows:
1) Although there are links to short videos in the material, it is suggested to extract several main categories for display in order to increase intuition.
2) Whether the data provided are marked with positive and negative samples and how to mark them.
3) The results given at present, such as how the distribution of fig1-3 is obtained, are suggested to be explained.
4) Does the public data provided involve personal privacy and ethical issues?
Author Response
We gratefully thank the reviewers for the comments that help improve the manuscript. Below the comments (in bold) are our responses point by point, and the manuscript has been revised accordingly. The revised text was in italics in this response letter and highlighted accordingly in yellow in the revised manuscript.
Comments and Suggestions for Authors
The research work and contribution of authors are to provide short videos for visual emotion research and build the corresponding database, which has certain value for the construction and testing of positive and negative samples in subsequent research.
Response: Thanks for the comments and the suggestions that help improve the manuscript.
My suggestions are as follows:
1) Although there are links to short videos in the material, it is suggested to extract several main categories for display in order to increase intuition.
Response: Thanks for the suggestions. We have added the category (people, animals, objects, and scenes) information for the emotion short-video library in Table S2 of supplementary materials, which is available at the same address as the video library: https://osf.io/rhuzy/files/.
2) Whether the data provided are marked with positive and negative samples and how to mark them.
Response: We have provided the information on how to mark the valence of the videos in the Methods of the revised manuscript,
“On the valence dimension, a video was determined to be negative emotion when scored less than 5, neutral when equal to 5, and positive when over 5. When videos are categorized according to the valence dimension, the recognition rate expects to be higher than 60.0% [36, 37]. That is to say, 60 out of 100 participants rated the video as the target emotion. For this reason, of the 299 videos, 57 videos whose ratings were lower than 60%were removed. Among the 242 selected videos, the numbers of videos across positive, neutral, or negative properties were 112, 47, and 83, respectively (Supplementary Table S1). The recognition rate and category information for the 242 out of 299 videos were listed in Supplementary Table S2. The average (across participants) score of each of the 242 videos from each emotional dimension (valence, arousal, and dominance) was shown in Table S3.”
3) The results given at present, such as how the distribution of fig1-3 is obtained, are suggested to be explained.
Response: We have added the following text in the Results section to explain how to obtain the distribution in Figures 1 to 3,
“We made several pairwise correlation analyses between the scores of the three emotion dimensions of valence, arousal, and dominance, obtaining the correlation distributions in Figures 1 to 3. When there was a nonlinear relationship in the scatter diagram, we conducted a three-dimensional correlation analysis for positive and negative videos separately. The single dot represents the average rating for one video on a two-dimensional scale, averaged across participants.”
4) Does the public data provided involve personal privacy and ethical issues?
Response: Our dataset contains videos of characters or faces. We have obtained permission from the persons appearing in the video or their guardians (such as a baby or child in the video clip) that the videos could be made public and used only for scientific research. For those who appear in the video with front faces, a signed paper-based informed consent before participating in the study was introduced, ensuring that they understood the study's purpose and agreed that the video should be made public and used only for scientific research.
Reviewer 2 Report
I have the following recommendations regarding improvements of the paper.
1. Add the motivation of the proposed approach.
2. Literature related to the proposed scheme published in the last 3 years should be added (at least 3 more references).
3. Write the Research gap i.e. which mentions the limitations in the literature and how the authors overcome it.
4. Problem statements need to be written that clearly illustrate which problem is the focus of the study.
5. Research contributions, which clearly describe the novelty of the proposed solution should be added in the form of bullets or numbered form (1, 2, 3, ...).
6. Mention some future directions and limitations of the proposed scheme in the conclusions section.
7. Mention all the acronyms in a table by the end of the manuscript.
8. Paper should be checked for english spellings/minor grammatical mistakes.
9. Improve the figures
Author Response
Reviewer 2
Comments and Suggestions for Authors
I have the following recommendations regarding improvements of the paper.
Response: We gratefully thank the reviewer for the comments that help improve the manuscript. Below the comments (in bold) are our responses point by point, and the manuscript has been revised accordingly. The revised text was in italics in this response letter and highlighted accordingly in yellow in the revised manuscript.
1. Add the motivation of the proposed approach.
Response: We have added the following text in the Methods and Results of the revised manuscript to provide the motivation of this study’s approach,
“2.4 Stimuli Presentation and On-line Experiment Procedure
This study was conducted in the spring of 2020. Due to the global COVID-19 pan-demic at that time, we chose online experimental research. We developed a website using JavaScript (version 2018) and nodeJS (version 4.4.3) for this experiment and stored visual clips on the cloud server. A user could log into the website with a unique assigned username and password to start the experiment.”
“2.5 Statistical Analysis
The statistical analyses were performed using the software SPSS 22.0. we first used descriptive statistics to show the sample characteristics under various conditions with-in-group or between-group, then independent sample t-tests to compare the mean difference of variables between gender groups, and then scatter charts and correlation analyses to show the distribution characteristics between two of the three dimensions (valence, arousal, and dominance) of emotion under various conditions.”
“3.5 Scatter Plots of Rating Distribution for Emotional Dimensions
We made several pairwise correlation analyses between the scores of the three emotion dimensions of valence, arousal, and dominance, obtaining the correlation distribution in Figures 1 to 3. When there was a nonlinear relationship in the scatter diagram, we conducted a three-dimensional correlation analysis for positive and negative videos separately. The single dot represents the average rating for one video on a two-dimensional scale, averaged across participants.”
2. Literature related to the proposed scheme published in the last 3 years should be added (at least 3 more references).
Response: Thanks for the suggestions. We have added the review of recent research related to our study in the revised Introduction section,
“Previous studies pointed out that the emotional intensity caused by dynamic expressions is higher than those of static ones [15]. Presently several video emotional stimulus databases are available. For example, some video libraries containing facial expressions and sounds have been developed and used worldwide [5, 16-20]. Of these face-emotion video libraries, there are datasets containing dynamic emotional faces performed by professional actors, such as the Geneva Multimodal Emotion Portrayal (GEMEP) corpus [16]. A library contains dynamic face expressions recorded by children, such as the Child Emotion Facial Expression Set (ChildEFES) [17]. In addition, some emotional video materials are from film clips, including characters and environmental scenes [21, 22].”
3. Write the Research gap i.e. which mentions the limitations in the literature and how the authors overcome it.
Response: Thanks for the suggestion. We have rewritten the last paragraph of the Introduction to clarify this study’s motivation,
“In real life, the generation of emotion mainly relates to dynamic simulation. Dynamic emotion videos have been widely used in social scene analysis [23] and emotion neural computing models or brain-like emotion recognition algorithms [24, 25]. The demand for stimuli with natural ecological validity in emotion recognition research is increasing, but due to the limitations of static stimuli and the content of previous video stimulus databases (i.e., the abovementioned databases include facial expressions or upper body movements only or clipped from non-commercial videos), it is necessary to expand the scope of dynamic visual stimulus materials to multiple categories of real natural scenes and character scenes in the research of emotion. Thus, in this study, to fill this research gap and provide more optional materials for researchers, we aimed to construct a short-video database containing various categories, including people, animals, objects, and natural and social scenes. We then tested the database among college students.
4. Problem statements need to be written that clearly illustrate which problem is the focus of the study.
Response: In the last paragraph of the Introduction section, we have pointed out the focus and aim of this study in the revised manuscript,
“ The demand for stimuli with natural ecological validity in emotion recognition research is increasing, but due to the limitations of static stimuli and the content of previous video stimulus databases (i.e., the abovementioned databases include facial expressions or upper body movements only or clipped from non-commercial videos), it is necessary to expand the scope of dynamic visual stimulus materials to multiple categories of real natural scenes and character scenes in the research of emotion. Thus, in this study, to fill this research gap and provide more optional materials for researchers, we aimed to construct a short-video database containing various categories, including people, animals, objects, and natural and social scenes. We then tested the database among college students.”
5. Research contributions, which clearly describe the novelty of the proposed solution should be added in the form of bullets or numbered form (1, 2, 3, ...).
Response: We have rewritten the strength of this study in the Discussion section of the revised manuscript as follows,
“The strength of this study includes: (1) The videos extended standard tools like CAPS by adding the dynamic dimension. (2) The removal of individuals with possible clinical depression or anxiety avoided the impact of the two factors on the emotion-rating of the videos. (3) The randomization of the order for playing the videos and balancing the participants’ gender ensured the standardization of the research. (4) From the perspective of the application prospect of the video library, future studies may find the application of the video library in other age groups and patient populations. Further, emotion videos have been used in recent social cognition studies, such as in the areas of visually-mediated empathy and somatosensory vicariousness and their interaction with neural activities [42, 43]. Dynamic visual affective stimuli are also valuable for emotional neutral computing models, such as recognizing personality [44] or distinguishing emotions [45] by emotion-al-video induced EEG signal. These application prospects imply that the visual emotion video library tested in this study, which has a standardized classification of the experimental stimuli, might help improve the design of human affective-oriented experiments and the accuracy of the neural computing predictive models.”
6. Mention some future directions and limitations of the proposed scheme in the conclusions section.
Response: Thanks for the suggestion. We have rewritten the Limitations of this study in the Discussion section,
“This study also has some limitations. First, the total number and category of the short-video stimuli should expand, especially those with clips in the high arousal/valence areas. Second, in the video evaluation, there was no test for the classification of complex emotions in the video, such as judging whether the positive emotions reflected in the video are joy or amusement and whether the negative emotions reflected in the videos are sadness or anger or fear. Future research can further improve this part and supplement the video as needed. Third, some social and natural scenes in the videos may have Oriental cultural characteristics, such as garden scenery and group dance, so their applicability in the cross-cultural field needs to be further verified.”
7. Mention all the acronyms in a table by the end of the manuscript.
Response: we have added the abbreviations by the end of the manuscript,
“Abbreviations
“BDI-II-C, Beck Depression Inventory-II in Chinese Version; CAPS, Chinese Affective Picture System; GAD-7, 7-items Generalized Anxiety Disorder Scale; IAPS, International Affective Picture System”
8. Paper should be checked for english spellings/minor grammatical mistakes.
Response: Thanks for pointing out the problem. We have checked the manuscript for English spelling and grammatical mistakes.
9. Improve the figures
Response: Thanks for the suggestion. We have reproduced the figures in the manuscript to ensure their clarity and informativeness.

Reviewer 3 Report
In the paper “Visual Affective Stimulus Database: A Validated Set of Short Videos” the authors investigated the emotional evaluation of affective videos. To this aim the authors asked 116 participants to rate 299 videos according to a 9-point rating scale. The study is sound and timely. The main findings showed that videos can extend standard tools like IAPS and CAPS by adding the dynamic dimension. This should be sufficient to warrant publication. However, there might be room for improving the discussion in order to better highlight the impact of the study in an extended framework. In particular, it could be worth discussing how the present study could help the development of affective-oriented experiments in humans and, therefore, contribute to extend the knowledge about key concepts of cognitive neuroscience and neuropsychology. For example, visual affective videos have been recently used to establish that visually-mediated psychological closeness interacts with the brain responsiveness to empathic processing (Ionta et al 2020, Cortex). In the same vein, visual affective videos have been proved efficient to assess that body ownership can modulate the brain activity associated with somatosensory vicariousness (Pamplona et al 2022, Cerebral Cortex). On this basis, it might be proposed that the implementation of the database examined by the present study in future affective-oriented could result in a more precise/standardized classification of the experimental stimuli, which in turn could provide a better focus on the cognitive neuroscience concepts (e.g. empathy, body ownership, vicariousness, etc) addressed by the specific experiments. This might offer a stronger background to show the impact of the present study in a broader framework. What do the authors think?
Author Response
Response: We sincerely appreciate the reviewer for the comments and suggestions. we have added the following texts in the Discussion to state the strength of this study,
“From the perspective of the application prospect of the video library, future studies may find the application of the video library in other age groups and patient populations. Further, emotion videos have been used in recent social cognition studies, such as in the areas of visually-mediated empathy and somatosensory vicariousness and their interaction with neural activities [42, 43]. Dynamic visual affective stimuli are also valuable for emotional neutral computing models, such as recognizing personality [44] or distinguishing emotions [45] by emotion-al-video induced EEG signal. These application prospects imply that the visual emotion video library tested in this study, which has a standardized classification of the experimental stimuli, might help improve the design of human affective-oriented experiments and the accuracy of the neural computing predictive models.”
Round 2
Reviewer 1 Report
The revised version has responded to all my previous concerns and can be published in the present form.
Reviewer 3 Report
accept